# Quorum Sensing Signaling Molecules Positively Regulate c-di-GMP Effector PelD Encoding Gene and PEL Exopolysaccharide Biosynthesis in Extremophile Bacterium *Acidithiobacillus thiooxidans*
[note 1]

**DOI:** 10.3390/genes12010069

**Published:** 2021-01-07

**Authors:** Mauricio Díaz, Diego San Martin, Matías Castro, Mario Vera, Nicolás Guiliani

**Affiliations:** 1Laboratorio de Comunicación Bacteriana, Departamento de Biología, Facultad de Ciencias, Universidad de Chile, Ñuñoa, Santiago 7800003, Chile; maujav2004@hotmail.com (M.D.); diego.sanmartin.f@ug.uchile.cl (D.S.M.); 2Fundación Ciencia y Vida, Ñuñoa, Santiago 7780272, Chile; mcastro@cienciavida.org; 3Instituto de Ingeniería Biológica y Médica, Escuelas de Ingeniería, Medicina y Ciencias Biológicas, Pontificia Universidad Católica de Chile, Macul, Santiago 7820436, Chile; maverav@uc.cl; 4Departamento de Ingeniería Hidráulica y Ambiental, Escuela de Ingeniería, Pontificia Universidad Católica de Chile, Macul, Santiago 7820436, Chile

**Keywords:** acidophile, biofilm, bioleaching, cyclic dinucleotide, epifluorescence microscopy, extremophile

## Abstract

*Acidithiobacillus* species are fundamental players in biofilm formation by acidophile bioleaching communities. It has been previously reported that *Acidithiobacillus ferrooxidans* possesses a functional quorum sensing mediated by acyl-homoserine lactones (AHL), involved in biofilm formation, and AHLs naturally produced by *Acidithiobacillus* species also induce biofilm formation in *Acidithiobacillus thiooxidans*. A c-di-GMP pathway has been characterized in *Acidithiobacillus* species but it has been pointed out that the c-di-GMP effector PelD and *pel*-like operon are only present in the sulfur oxidizers such as *A. thiooxidans*. PEL exopolysaccharide has been recently involved in biofilm formation in this *Acidithiobacillus* species. Here, by comparing wild type and Δ*pelD* strains through mechanical analysis of biofilm-cells detachment, fluorescence microscopy and qPCR experiments, the structural role of PEL exopolysaccharide and the molecular network involved for its biosynthesis by *A. thiooxidans* were tackled. Besides, the effect of AHLs on PEL exopolysaccharide production was assessed. Mechanical resistance experiments indicated that the loss of PEL exopolysaccharide produces fragile *A. thiooxidans* biofilms. qRT-PCR analysis established that AHLs induce the transcription of *pelA* and *pelD* genes while epifluorescence microscopy studies revealed that PEL exopolysaccharide was required for the development of AHL-induced biofilms. Altogether these results reveal for the first time that AHLs positively regulate *pel* genes and participate in the molecular network for PEL exopolysaccharide biosynthesis by *A. thiooxidans*.

## 1. Introduction

The biomining industry takes advantage of the metabolism of leaching microorganisms which mediate the dissolution of metal sulfides through their ability to oxidize ferrous iron and reduced inorganic sulfur compounds (RISCs). Bioleaching has been successfully used for the recovery of cobalt, gold, nickel, zinc and it is currently used for the recovery of copper from low-grade ores [1,2]. However, the leaching of metal sulfides under uncontrolled circumstances creates environmental pollution in the form of acid mine/rock drainage (AMD/ARD) [3]. It has been reported that biofilm formation by bacterial cells on minerals is a key step for leaching performance due to the formation of a thin reaction space between ore and cells, which is filled by extracellular polymeric substances including exopolysaccharides (EPS), proteins, lipids and uronic acids [4,5]. Understanding the molecular events involved in biofilm formation by acidophilic species may help to develop improvements in biomining technologies or to mitigate AMD/ARD. Due to their role in bioleaching, bacteria belonging to *Acidithiobacillus* genus were the first acidophilic species to be characterized and considered to be pivotal players for the biomining process [6,7].

In bacteria, biofilm formation is mainly controlled by two specific and widespread phenomena named quorum sensing (QS) [8,9] and the cyclic diguanylate (c-di-GMP) pathway [9,10,11,12]. QS mechanisms have been studied for several decades in Gram-negative and Gram-positive bacteria and in addition they also have been identified in eukaryotic fungi and the Protista kingdom [13,14]. QS is defined as a cell-cell communication process that regulates gene expression in a cell-density-dependent manner. This is achieved through the secretion of diffusible autoinducers (AIs), allowing the expression of different behaviours more suitable for the cell population rather than individual cells [8]. These include virulence, EPS biosynthesis and biofilm formation. Different QS systems have been reported, and characterized AIs have been described as specific molecular players for intraspecies as well as interspecies cell-cell communication [15,16,17,18,19]. Canonical QS systems in Gram-positive bacteria involve oligopeptides as AI and a specific two-component system for the signal transduction [20]. QS type AI-1, which is mediated by *N*-acyl-homoserine lactone (AHL) molecules through its binding to transcriptional regulators belonging to the LuxR-like protein family, is one of the best characterized QS systems in Gram-negative bacteria [20]. A type AI-1 QS system was reported in *Acidithiobacillus ferrooxidans* [21]. It includes canonical *afeR* and *afeI* genes, encoding for the transcriptional regulator AfeR and the AHL synthase AfeI. Since AfeI is capable of driving the biosynthesis of nine different AHLs, it has been suggested that cross-communication could occur between *A. ferrooxidans* and other bacterial species belonging to the bioleaching ecological niche [22]. In this way, it has been reported that attachment to pyrite by the sulfur oxidizer *Acidithiobacillus thiooxidans* requires a pre-colonization step mediated by iron oxidizing species [23]. In addition, the use of synthetic QS molecules revealed that AHLs naturally produced by iron/sulfur oxidizer *A. ferrooxidans* promote biofilm formation not only in *A. ferrooxidans* [24], but also in the sulfur oxidizer *A. thiooxidans* [23]. An initial bioinformatics analysis suggested that the *A. ferrooxidans* QS regulon may comprise 75 genes some of them involved in polysaccharide biosynthesis [25]. Moreover, DNA microarray experiments performed in *A. ferrooxidans* by using an AHL super-agonist analog indicated that 42.5% of predicted QS regulon genes are related to biofilm formation and to the biosynthesis, polymerization and secretion of exopolysaccharides [26]. Nevertheless, the molecular network involved in biofilm formation by *Acidithiobacillus* species, from the addition of exogenous AHL to EPS production upon cell attachment on pyrite or sulfur surfaces, remains to be deciphered.

The second messenger c-di-GMP has emerged as a central metabolite that controls several phenotypes in bacteria, including motility and biofilm formation [11,12]. This messenger is synthesized by diguanylate cyclase (DGC) enzymes and degraded by c-di-GMP specific phosphodiesterases (PDEs) [11,12]. The signal transduction is carried out by several families of proteins and RNA receptors [27]. One of the first c-di-GMP effector proteins to be characterized was the inner-membrane protein PelD from *Pseudomonas aeruginosa* [28]. The PelD protein is part of a multiproteic complex involved in the biosynthesis and export of PEL, an exopolysaccharide involved in pellicle formation at the air/surface interface of *P. aeruginosa* static liquid cultures [28]. The unique architecture and export mechanism of the PEL polysaccharide synthase, as well as the structural composition of PEL exopolysaccharide have recently been deciphered in *P. aeruginosa* [29,30,31]. PEL is a cationic exopolysaccharide, mainly composed of *N*-acetylgalactosamine and *N*-acetylglucosamine subunits [29], whose translocation across the outer membrane requires functional PelB and PelC proteins [30], and the binding of c-di-GMP to PelD for recruiting PelF and promoting its glycosil-transferase activity through quaternary rearrangements of the PEL polysaccharide synthase PelDEFG [31,32]. Functional c-di-GMP pathways have been reported and partially characterized in three *Acidithiobacillus* species, and directly related to exopolysaccharide production and biofilm formation [33,34,35]. In addition, by analysing 35 chromosomal replicons [36] it has been recently reported that the c-di-GMP pathway is widespread among the *Acidithiobacillus* species complex. Castro et al. [36] also corroborated that: (i) the c-di-GMP network is highly diverse, depending on both species and strains, the most complex c-di-GMP metabolism pathways being identified in *A. thiooxidans* strains, which can harbour up to 40 DGC and PDE encoding genes [36]; (ii) despite a wide diversity of c-di-GMP effectors such as the transcriptional regulator FleQ [37] and PilZ domain, initially identified in the cellulose synthase subunit BcsA [38], PelD and the *pel* operon were identified in very few species of acidophile. Thus, two operons, encoding for cellulose (*bcs* operon) and PEL exopolysaccharide (*pel* operon) biosynthetic pathways have been suggested to be involved in biofilm formation in *Acidithiobacilus* species [34,35]. However, while the *bcs* operon is widespread in iron/sulfur-oxidizing species, the *pel* operon [39] has been identified only in the sulfur-oxidizing species *A. caldus* and *A. thiooxidans* [30,34,35]. Indeed, a *pel*-like operon was identified in *A. caldus* and *A. thiooxidans*, and the construction of an *A. thiooxidans* Δ*pelD* null-mutant strain revealed that PEL exopolysaccharide is involved in its biofilm architecture [35]. Therefore, PEL was the first exopolysaccharide experimentally linked with biofilm formation by *Acidithiobacillus* species [35].

During the last decade, several reports have revealed that QS and the c-di-GMP pathway form intricate molecular networks that can integrate data on population density and environmental conditions in different bacterial species [40,41,42,43,44]. Indeed, pellicle formation at the air–surface interface of a bacterial culture, which is supported by the production of PEL and BEP exopolysaccharides in *P. aeruginosa* and *Burkholderia cenocepacia*, respectively, is induced by high intracellular levels of c-di-GMP, but it is negatively regulated by two different QS pathways decreasing c-di-GMP content either deactivating DGC or activating PDE enzymes [40,42,44,45]. However, despite the results reported by Diaz et al. [35], the identification of the molecular players involved in biosynthesis of PEL exopolysaccharide and biofilm formation by *Acidithiobacillus* is still mostly incomplete. In order to gain insights into the regulatory network involved in the biosynthesis of PEL exopolysaccharide by *Acidithiobacillus* species, and also to assess if QS signalling, c-di-GMP and PEL exopolysaccharide are interconnected during biofilm formation by *A. thiooxidans* ATCC 19377^T^, fluorescence microscopy and qPCR experiments were performed, taking advantage of the Δ*pelD* null-mutant strain [35]. The results of the present work demonstrated that in our experimental conditions transcription levels of *pel* genes were increased when *A. thiooxidans* planktonic cells were exposed to QS signalling molecule 3-oxo-C8-AHL, while fluorescent lectin binding analysis (FLBA) combined with epifluorescence microscopy clearly indicated that PEL exopolysaccharide was required for the development of AHL-induced biofilms. Therefore, this work provides the first evidence that QS signalling molecules are positively linked to the transcription of *pel* genes including the c-di-GMP effector encoding gene *pelD* and PEL exopolysaccharide biosynthesis during biofilm formation in *Acidithiobacillus* species. Nevertheless, further studies are still necessary to fully decipher the interplay of QS and c-di-GMP molecular pathways in *Acidithiobacillus* species.

## 2. Materials and Methods

### 2.1. Strains and Growth Conditions

*A. thiooxidans* ATCC 19377^T^ parental strain and null-mutant Δ*pelD* derived strain whose biofilm architecture is modified by this deletion [35] were routinely grown with agitation (150 rpm) at 30 °C under aerobic conditions in Mackintosh (MAC) medium at pH 4.5 [46], supplemented with 5% *w*/*v* sulfur (S°) prills as energy substrate. For fluorescence microscopy experiments and mechanical strength analysis, S°-coupons (0.5 cm^2^, obtained by S° melting and fusion) were added to the MAC medium. The null-mutant strain Δ*pelD* was maintained in MAC medium supplemented with 150 μg/mL kanamycin. For *A. thiooxidans* growth in the presence of AHL signalling molecules (5 µM final concentration), 3-oxo-C8-AHL and C8-AHL (SIGMA^®^, Oakville, ON, Canada) were added from 50 mM stock solutions in 100% DMSO.

### 2.2. Visualization of A. thiooxidans^T^ Biofilms

Colonized S°-coupons obtained from three independent cultures of *A. thiooxidans*^T^ cells grown in the presence of 3-oxo-C8-AHL (5 µM), C8-AHL (5 µM) or 0.01% DMSO were extracted from 5-days growth cultures and washed as described [35]. FLBA was done as described by Zhang et al. [47]. Coupons were then incubated in darkness for 1 h in lectin buffer (10 mM NaH_2_PO_4_ pH 7.2, 150 mM NaCl), supplemented with different FITC-conjugated lectins at 50 µg/mL (Appendix A) (EY Laboratories^®^, San Mateo, CA, USA). For FITC-conjugate GS-II lectin, CaCl_2_ (0.5 mM) was added into the lectin buffer. After incubation, S°-coupons were washed with the same lectin buffer and counterstained with 4,6-diamidino-2-phenylindole (DAPI) at 1 mg/mL in 2% formaldehyde, to fix the samples [23]. Finally, coupons were washed with sterile water, dried at room temperature, mounted with a drop of an anti-fading agent (Citifluor^®^ AF2) and imaged by epifluorescence microscopy. Images were taken with an inverted Axiovert-100 MBP microscope (Zeiss^®^) equipped with an HBO 100 mercury vapour lamp, filters for DAPI (Ex 358 nm/Em 461 nm) and FITC (Ex 490 nm/Em 505–545), a Zeiss^®^ filterset 49 air-objective (Zeiss^®^ EC epiplan NEOFLUAR 420363/9901) and a digital microscope camera (Zeiss^®^ AxioCam^®^ MRm). All images were acquired by viewing five different coupons from each independent culture and processed using the software Axio-Vision 4.2 (Zeiss^®^).

### 2.3. Transcriptional Analyses

Real Time RT-PCR experiments (qRT-PCR) were performed as previously described [35]. Total RNA was extracted from both planktonic and biofilm cells obtained from *A. thiooxidans*^T^ cultures grown in the presence of 3-oxo-C8-AHL (5 µM) or 0.01% DMSO during five days. Planktonic cells were directly collected by centrifugation for 15 min at 6000× *g* while biofilm cells were previously separated from S°-prills by 10 min incubation with 0.05% Triton X-100 and vortexing [35]. cDNA was synthetized from 1 µg of total RNA obtained from both cell sub-populations by using reverse transcriptase (Promega, Madison, WI, USA) and random primers (Promega, Madison, WI, USA). Transcriptional levels of *pel* (*pelA*, *pelD* and *wcaG*) and *flaA* genes were measured with specific primers (Appendix A), including both *16S rDNA* gene and the methionine aminopeptidase encoding gene *map*, used as housekeeping genes for data normalization [48].

### 2.4. Mechanical Resistance of A. thiooxidans Biofilms

To investigate the role of both AHL signalling molecules and PEL exopolysaccharide on biofilm formation by *A. thiooxidans*^T^, a specific assay to measure mechanical resistance of attached cells was developed. First, to eliminate any remaining planktonic cells, colonized S°-coupons were extracted from three independent 5-days growth cultures of *A. thiooxidans*^T^ cells and washed twice with aqueous H_2_SO_4_ pH 2. The mechanical resistance of attached cells was assessed by incubating the colonized S°-coupons in a Triton X-100 solution (0.05% Triton X-100, pH 2) and vortexing for 10 min. Afterwards, the *A. thiooxidans* cells released were quantified at different times of incubation by separating free *A. thiooxidans* cells from colonized S°-coupons through centrifugation at low velocity (1000× *g*). Suspensions of released cells were diluted in acidic water (pH 2.0, H_2_SO_4_) and counted in a Petroff-Hausser counting chamber. The number of released cells was normalized by the mass of S°-coupons.

### 2.5. Bioinformatics Search for a LuxR-Like Protein

The search for a LuxR-like orthologs in *A. thiooxidans* proteomes was performed by Blastp [49] with defaults parameters (*e*-value < 0.05) using AfeR (*A. ferrooxidans*), SdiA (*Escherichia coli*) and RpaR (*Rhodopseudomonas palustris*) as queries against the new available genome sequences of *A. thiooxidans*. Functional protein domains from queries and subject proteins were validated against the Conserved Domain Database (CDD) v.3.18 and Pfam Database V.32.0 using CD-search [50] with defaults parameters (*e*-value < 0.01).

## 3. Results

### 3.1. N-Acetyl-Galactosamine and N-Acetyl-Glucosamine Are Structural Blocks of PEL Exopolysaccharide in A. thiooxidans^T^

To gain some insights about the sugar composition of the *A. thiooxidans* cell surface and to understand how PEL exopolysaccharide is involved, FLBA was performed using twelve different lectins. In our experimental conditions, most of the tested lectins did not bind *A. thiooxidans* wild type cells (Appendix A). However, binding signals were observed with lectins AAL, BPA, ConA and GS-II, which bind l-Fucose α(1,6) *N*-Acetyl-d-Glucosamine, *N*-acetyl-d-galactosamine, internal d-mannose and d-Glucose and *N*-acetyl-d-glucosamine, respectively (Figure 1 and Appendix A).

In agreement with Diaz et al. [35], DAPI staining corroborated that wild type and Δ*pelD* null-mutant cells are capable of forming biofilms on S°-coupons (Figure 2 and Appendix A) and definitively pointed out that *A. thiooxidans* cells can also adhere to S°-coupons independently of PEL. AAL, BPA and GS-II clearly indicated that the glycoconjugate composition of biofilm cell surfaces is different in the two strains. Positive binding signals were obtained with the aforementioned three lectins for the wild type strain, while no (BPA, GS-II) or decreased (AAL) fluorescence signals were obtained for Δ*pelD* null-mutant strain (Figure 2 and Appendix A), indicating that the PEL exopolysaccharide of *A. thiooxidans* is most likely composed of *N*-acetyl-d-galactosamine and *N*-acetyl-d-glucosamine.

### 3.2. The Loss of PEL Exopolysaccharide Produces Fragile Biofilms in A. thiooxidans

Diaz et al. [35] have previously reported that Δ*pelD* mutation affects biofilm architecture. In order to assess the relevance of PEL exopolysaccharide for *A. thiooxidans* biofilm architecture and function, wild type and Δ*pelD* null-mutant biofilms were subjected to mechanical stress by vortexing colonized S°-coupons. As shown in Figure 2, the number of cells released at the end of vortexing (10 min) from control Δ*pelD* biofilms without 3-OH-C8-AHL was 29.9% higher than those released from wild type biofilms. This increase was also observed at different times of the vortexing assay (Appendix A).

Concordantly, and compared to the Δ*pelD* null-mutant strain, the presence of PEL exopolysaccharide reduced the release of wild type cells from S°-coupons in presence of the QS signalling molecule 3-oxo-C8-AHL by 53.3% (Figure 2). Moreover, a significant decrease (26.4%) was also observed for WT cells grown with 3-oxo-C8-AHL compared to WT cells without QS signalling molecules and the highest releases were observed for Δ*pelD* biofilms with or without 3-oxo-C8-AHL (Figure 2). Altogether, these results clearly show that the strength of *A. thiooxidans* cells embedding and the mechanical resistance of the biofilm matrix are correlated to the presence of PEL exopolysaccharide. In addition, our results highlight for the first time a role for an AHL signalling molecule in PEL biosynthesis by this acidophile species.

### 3.3. PEL Biosynthesis by A. thiooxidans Requires QS Signalling Molecules

Ueda and Wood [40] reported that QS mediated by the Las system negatively modulates pellicle/biofilm formation in *P. aeruginosa*. In addition, while sulfur-oxidizing species such as *A. thiooxidans* and *A. caldus* do not possess any canonical genes for QS [51,52], it has been reported that biofilm formation in *A. thiooxidans*^T^ can be induced by the addition of QS signalling molecules 3-oxo-C8-AHL or C8-AHL [23]. Thus, we decided to assess the influence of the addition of these AHLs on PEL production and biofilm formation, by comparing the *A. thiooxidans* Δ*pelD* null-mutant and WT strains. Despite the basal level of adhered cells observed in both controls without QS signalling molecules and all Δ*pelD* experiments that corroborated that *A. thiooxidans* can adhere to S°-coupons independently of PEL, results obtained by fluorescence microscopy clearly indicated that the formation of the AHL-induced biofilm by *A. thiooxidans* requires the presence of the c-di-GMP effector PelD (Figure 3).

This indicates that these QS signalling molecules have a positive effect on PEL biosynthesis and suggests that it could regulate the expression of the PEL apparatus. In order to test this hypothesis, transcriptional analyses of three genes belonging to the *A. thiooxidans pel* operon (*pelA*, desacetylase PelA; *pelD*, c-di-GMP effector protein PelD; *wcaG*, UDP-Glucose-4-epimerase) were performed in the presence of 3-oxo-C8-AHL (5 μM), since it appeared to induce the highest biofilm formation in this microorganism (Figure 3). No difference was measured for *pelA, pelD* and *wcaG* transcription levels by comparing biofilm cells with or without addition of 3-oxo-C8-AHL (Figure 4 and Appendix A). However by comparing data obtained from planktonic and attached cells in controls (CTR) experiments, it is possible to point out that basal transcription levels are higher in biofilm cells compared to planktonic. This supports the idea that AHL effect on *pelA* and *pelD* transcriptions takes place early in *A. thiooxidans* planktonic cells to promote the shift of lifestyle from planktonic to adhered as it was reported by a global transcriptional analysis in *A. ferrooxidans* performed to assess the transcriptional effect of an AHL analogue on biofilm formation by this *Acidithiobacillus* species [26].

However, in planktonic cells, the transcription levels of the *pelA* and *pelD* genes were increased 2.2- and 2.8-fold in presence of 3-oxo-C8-AHL (5 μM), compared to control assays without its addition (Figure 4), while no significant change was observed for the last gene of the *pel* operon, *wcaG* (Appendix A). Because it has been reported that motility and biofilm are two phenotypes regulated in an opposite manner by intracellular levels of c-di-GMP, the transcription levels of *flaA,* a flagellin-like encoding gene present in *A. thiooxidans* ATCC 19,377 genome, was also assessed. However, none differences were observed by comparing WT and Δ*pelD* null-mutant planktonic cells grown with or without AHL (Appendix A).

## 4. Discussion

PEL was recently identified as a structural exopolysaccharide for biofilm formation by *A. thiooxidans* and it was reported that biofilm structures are different in *A. thiooxidans* WT strain compared to Δ*pelD* null-mutant strain that overexpressed a filamentous appendix [35]. However, a better characterization of the molecular events involved in the regulation of PEL biosynthesis is still required to understand its role in biofilm formation and architecture of these acidophilic species. To address this question, the Δ*pelD* null-mutant strain developed by Diaz et al. [35] was used to analyse the effect of QS signalling molecules, extracellular glycoconjugate diversity, mechanical resistance of biofilms, and to gain more precise new insights into the molecular network involved in the regulation of PEL exopolysaccharide biosynthesis in *Acidithiobacillus* species that can only oxidize RISCs.

In agreement with results obtained for *P. aeruginosa* [29], PEL exoplysaccharide from *A. thiooxidans* appears to be mainly composed of *N*-acetyl-d-galactosamine and *N*-acetyl-d-glucosamine. The presence of an additional *wcaG* gene encoding for an UDP-glucose-4-epimerase [35], downstream of the canonical *pel* operon, which is overexpressed in biofilm cells compared to planktonic cells (Appendix A), strongly suggested that the formation of UDP-*N*-acetyl-galactosamine from UDP-*N*-acetyl-glucosamine could be catalysed by this enzyme, as it has been recently reported by Whitfield et al. [31]. These data point out that the biofilm architecture differences observed between wild type and Δ*pelD* null-mutant biofilms recently reported by Diaz et al. [35] can be directly related to the presence of PEL exopolysaccharide and its main structural components *N*-acetyl-d-galactosamine and *N*-acetyl-d-glucosamine.

Altogether, the results obtained here strongly suggested that the QS molecules 3-oxo-C8-AHL and C8-AHL positively regulate PEL biofilm production by increasing the transcription levels of PEL-apparatus encoding genes including the c-di-GMP binding protein PelD. Interestingly, this result differs with the work of Ueda and Wood [40], which reported that QS negatively regulated PEL production in *P. aeruginosa* by decreasing c-di-GMP biosynthesis, but it agrees with Pérez-Mendoza et al. [53] who revealed that the ExpR/SinI QS system mediated by AHLs positively regulates the transcription of the *bgsA* gene encoding for a c-di-GMP effector protein involved in the synthesis of MLG exopolysaccharides by *Sinorhizobium meliloti*.

On the other hand, qPCR results reported in this work also revealed that transcription levels of *flaA* do not change in *A. thiooxidans* WT and Δ*pelD* cells grown with or without AHL molecules. Then the hypothesis suggesting that the overexpressed filamentous appendix observed in biofilm produced by *A. thiooxidans* Δ*pelD* strain could correspond to a mesh of entangled flagella [35] can be discarded. The ability of the *A. thiooxidans* Δ*pelD* cells to still be adhered on S°-coupons could be related to the presence of the *bcs* operon identified in the genome sequence of *A. thiooxidans*^T^ [34,35] and then cellulose could be responsible for this AHL independent adherence.

How the addition of AHLs increases *pel* genes mRNAs and enhances attachment to surfaces is a matter of discussion and is still an open question as illustrated by Figure 5.

Recently, it has been proposed that the flexibility to QS-signalling molecules such as AHLs is mainly due to variability of AHL-receptor proteins [54]. Valdés et al. [51,52] reported that sulfur-oxidizing species *A. thiooxidans* and *A. caldus* do not possess any canonical genes for QS system. However, it is well established now that *A. thiooxidans* is capable to sense AHL signalling molecules to induce biofilm formation [23] (this work) This indicates that an unknown AHL-receptor protein has to exist in this acidophile species and may act as a transcriptional regulator either directly to promote the transcription of *pel* genes (Figure 5, #3a), or indirectly to induce the transcription of DGC and PDE encoding genes involved in c-di-GMP metabolism (Figure 5, #3c) but also c-di-GMP protein receptors encoding genes such as the transcriptional regulator FleQ (Figure 5, #3b). Further studies are necessary to identify and characterize this unknown AHL-receptor protein. Nevertheless, by performing a bioinformatics analysis on the newly available *A. thiooxidans* genome sequences (strains A01, CLST, DXS-W, BY-02, A02, DMC, GD1-3, JYC-17, and ZBY) in which a *pel* operon has been identified (personal communication), we have recently identified a SdiA-like protein with a high *e*-value (Appendix A) that appears as a strong candidate to play the molecular role of QS transcriptional regulator. Effectively, SdiA is an orphan QS transcriptional regulator, not associated with an AHL synthase, that binds AHLs and has been related to motility and biofilm formation in *Escherichia coli* and *Salmonella enterica* serovar Thyphimurium [55,56]. Interestingly, Prescott and Decho [54] have proposed the orphan QS transcriptional regulators as key molecular players for *flexibility and adaptability of QS,* especially to maintain or develop cell-cell communication during the dynamic evolution of a biofilm community. Indeed, a bioinformatic analysis of the new available genome sequence on ten different genome sequences also revealed that *A. thiooxidans* and *A. caldus* possess several copies of the *fleQ* gene [57]. Then, a possible mechanism of action of the 3-oxo-C8-AHL could be promotion of transcription of a FleQ encoding gene, which in turn could induce PEL exopolysaccharide biosynthesis, as occurs in *P*. *aeruginosa* [37]. The existence of interplays between QS and c-di-GMP pathways has been demonstrated in other bacterial species [40,41], in which certain QS molecules regulate activity levels of DCG and PDE enzymes. Therefore, we can also hypothesize that in *A. thiooxidans* some of the genes targeted by the binary complex (3-oxo-C8-AHL/transcriptional regulator) may encode for proteins with DGC and/or PDE activities, producing an increase in intracellular c-di-GMP levels and consequently PEL-biofilm formation.

## 5. Conclusions

Here we report that PEL-biofilm contributes to *A. thiooxidans* cells resistance to mechanical stress. We also reveal that the positive effect induced by QS signalling molecules on biofilm formation by *A. thiooxidans* is directly mediated by PEL exopolysaccharide. We highlight that AHL molecules induce the transcription of several genes belonging to the *pel* operon including *pelD*, which encodes the c-di-GMP effector protein PelD. Finally, these results offer the first opportunity to propose a working model (Figure 5) that will enable further molecular characterization of the regulating network involved in PEL-biofilm formation by obligate sulfur-oxidizing *Acidithiobacillus* species.

## Figures and Tables

**Figure 1 genes-12-00069-f001:**
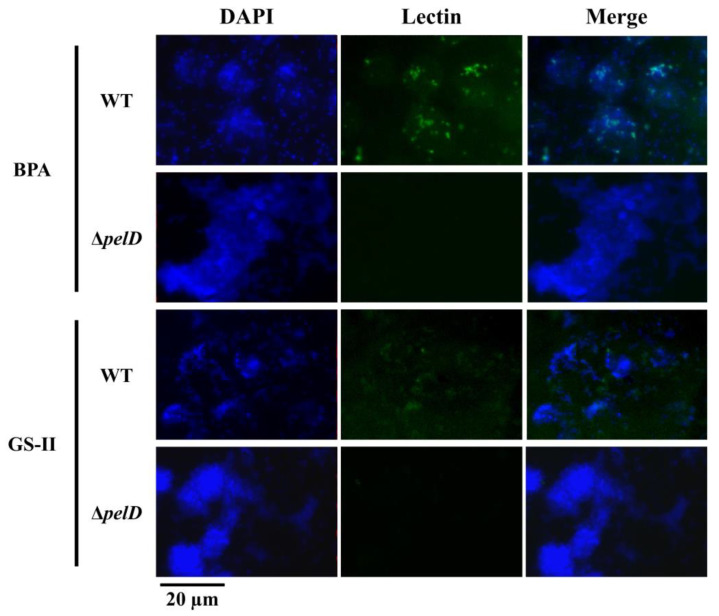
Analysis of PEL exopolysaccharide composition by using epifluorescence microscopy coupled to FLBA. S°-coupons colonized by *A. thiooxidans*^T^ (WT) or mutant derived (Δ*pelD*) cells were extracted from 5-days growth cultures and incubated with FITC-conjugated BPA or GS-II lectins. Then, they were stained with DAPI before microscopy imaging. Size bars represent 20 µm.

**Figure 2 genes-12-00069-f002:**
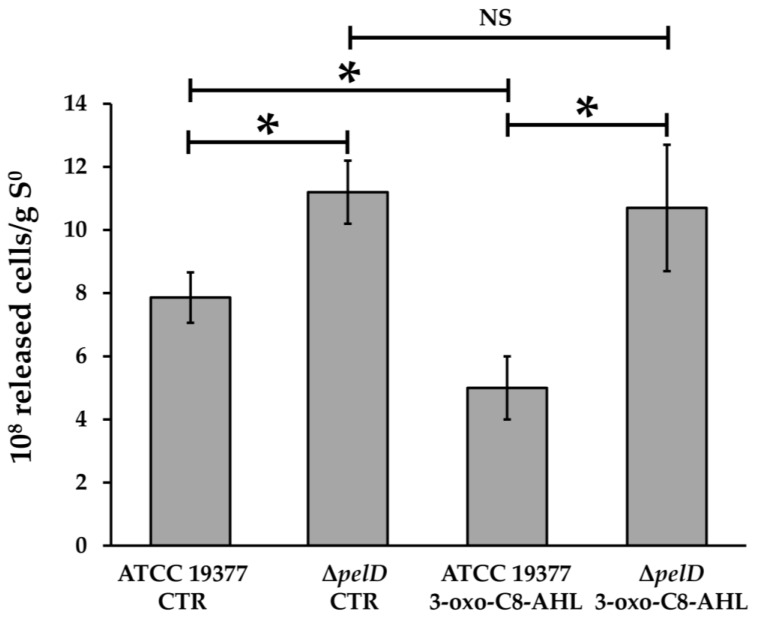
The presence of PEL exopolysaccharide offers a stronger embedment into the biofilm matrix for *At. thiooxidans* cells. Inoculated S°-coupons extracted from 5-days growth cultures were treated with 0.05% Triton X-100 and vortexed during 10 min. Number of cells released from the wild type and ΔpelD null-mutant biofilms subjected to mechanical stress was determined with a Petroff-Hausser counting chamber and normalized against mass of sulfur. Significant differences calculated by a one-way ANOVA test (*p* < 0.05) are noted (*). CTR, DMSO 0.01% without AHL. NS, No significant difference.

**Figure 3 genes-12-00069-f003:**
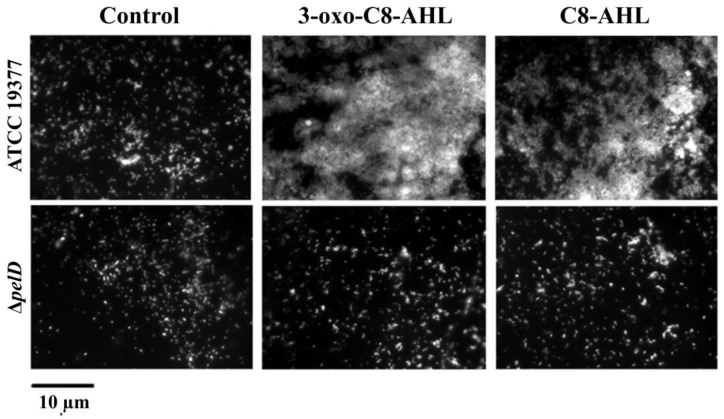
The deletion of *pelD* interferes with *A. thiooxidans* biofilm response to both QS signalling molecules 3-oxo-C8-AHL and C8-AHL. S°-coupons were inoculated with *A. thiooxidans* ATCC 19,377 or mutant derived Δ*pelD* strains and extracted from 5-days growth cultures. Then they were washed and stained with 0.01% DAPI. Finally, S°-coupons were viewed by fluorescence microscopy. Size bars represent 10 µm. Control, DMSO 0.01% without AHL.

**Figure 4 genes-12-00069-f004:**
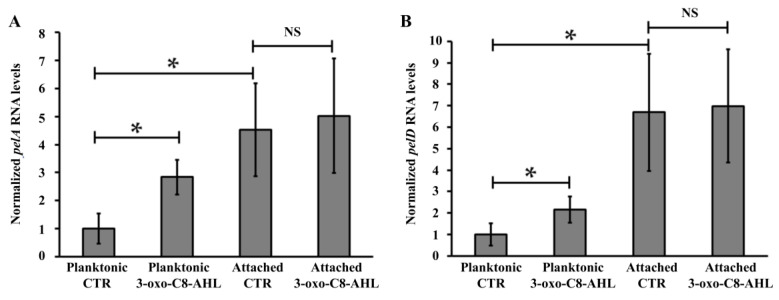
Effects of 3-oxo-C8-AHL addition on transcription levels of *pelA* and *pelD* genes. Transcript levels of *pelA* (**A**) and *pelD* (**B**) genes were measured by qRT-PCR. Total RNAs were obtained from 5-days growth cultures. Data were normalized using DNA *16S* and *map* genes. Values represent the average of 4 independent experiments ± standard deviation. Significant differences calculated by a one-way ANOVA test (*p* < 0.05) are noted (*). CTR, DMSO 0.01% without AHL. NS, No significant difference.

**Figure 5 genes-12-00069-f005:**
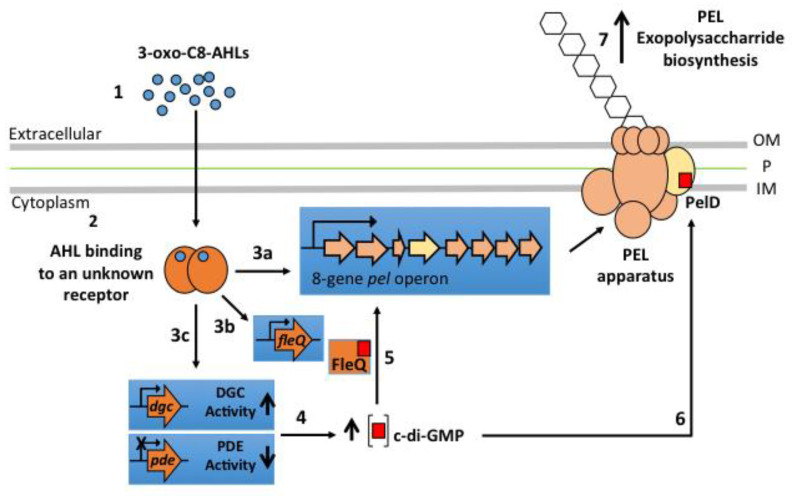
Working model for the regulation of PEL exopolysaccharide biosynthesis by obligate sulfur-oxidizing *Acidithiobacillus* species based on experimental data previously reported and obtained here with *A. thiooxidans*. The binding of 3-oxo-C8-AHL (1) by an unknown AHL receptor that could act as a positive transcriptional regulator (2) can directly promote *pel* operon expression (3a; Figure 3). But it may also induce the transcription of FleQ (3b) and diguanilate cyclase (DGC) (3c) encoding genes and/or repress the transcription of phosphodiesterases (PDE) encoding genes (3c). The new balance between DGC and PDE activities generate an increase of intracellular c-di-GMP levels (4). Then, c-di-GMP can bind to its specific receptors FleQ (5) and/or PelD (6) promoting the biosynthesis of PEL exopolysaccharide (7). OM, Outer membrane; P, Peptidoglycan; IM, Inner membrane.

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
