# Peer review of "Quorum Sensing Signaling Molecules Positively Regulate c-di-GMP Effector PelD Encoding Gene and PEL Exopolysaccharide Biosynthesis in Extremophile Bacterium *Acidithiobacillus thiooxidans"

_genes, 2021, doi:10.3390/genes12010069_

Round 1

Reviewer 1 Report

This study examined the impact of the loss of PEL polysaccharides on the ability of Acidithiobacillus species to form biofilms.  Loss of PEL resulted in more fragile biofilms.  The study also found that quorum sensing by acyl-homoserine lactones positively regulated pel genes and are part of a network for biofilm biosynthesis.

Biologically induced acid mine drainage is a serious problem occurring in mines containing sulfide ores.  Previous reports indicate that biofilm formation may be a key step resulting in the production of environmentally damage acids. 

There is considerable study of biofilm formation associated with medical devices.  Some of this knowledge would have improved the experimental techniques used.  Vortexing coupons to determine how many cells are dislodged is not a very quantifiable way to determine mechanical film strength.  How do the authors know that they are vortexing the same every time?  While it is noted that there were statistically significant differences in results, the error bars were quite large and did overlap in some cases.  Medical biofilm literature discusses many ways to consistently assess film strength and these should have been adopted.

This paper builds upon previous work done by the group to study the impact of deletion of the pel genes.  It did show that deletion of these genes results in differing biofilm composition.  However, it would have been nice to have a discussion of what that meant.  Do the differing polysaccharides have different strengths, do those polysaccharides not adhere to each other as well?  What is the structure of the biofilms?  The results for the transcription three genes hypothesized to have a positive effect on PEL biosynthesis were inconclusive and did not show any statistical differences although it was stated in the discussion that there were differences.

The text is clear and easy to read. The paper is reasonably well written although an English editor would help.

The transcription results are overstated and the experimental design needs work. 

They address the main question posed. While interesting, this work does not seem that novel. This paper is incremental over the group’s previously published results. In my opinion, that paper could be published as a note at best.  Intriguingly, the authors discuss bioinformatic analyses that ID a possible SdiA protein that could be an unknown AHL molecule.  If these data were added to this paper, it would be much stronger and would be a candidate for a full paper.  The mechanical biofilm assessment should also be redone with better methods, looking to medical biofilm literature.

Reviewer 2 Report

This manuscript represent an extension of Authors’ previous work on Acidithiobacillus biofilms. Understanding of biofilm formation by acidophilic species is an important step for the development of biomining technologies.

The authors have demonstrated:

  1. That biofilms made by ΔpeLD strain are more fragile compared to WT strain biofilms
  2. Based on experiments with different kinds of lectins, exopolysaccharides in thiooxidans biofilms are composed of N-acetyl-D-galactosamine and N-acetyl-D-glucosamine
  3. Biofilm formation is induced by the addition of QS molecules, 3-oxo-C8-AHL and C8-AHL in the WT strain. On the other hand, in the ΔpelD strain this process is impaired, as shown by fluorescence microscopy experiments.
  4. Expression of pel genes is induced in planktonic cells by the addition of 3-oxo-C8-AHL molecule, while it has no effect on expression of same genes in the attached cells.

The work is generally solid, story is coherent and easy to follow, however some improvement are needed to be taken into consideration for the publication in Genes.

Comments:

  • The authors proved that deletion of pelD gene has an effect on biofilm formation and quality. However, from Figure 2 we see that it is still capable of biofilm formation, and not just attachment as demonstrated in Figure 3. It would be informative if time of culture growth and time points of RNA isolation is indicated to clear out when this biofilm is formed.
  • Higher expression of pel genes (pelA, pelD, wcaG) in planktonic cells compared to attached cells was demonstrated before by the same authors (Diaz et al., 2018.). There is an increase of expression of pelA, pelD and wcaG mRNA upon treatment with AHL only in planktonic cells. Contrary, AHL had no effect on pel genes expression in the attached cells, since basal expression was already at high level. To make firm conclusion on the effect of AHL on expression of pel genes and consequent formation of biofilms, author should provide evidence that planktonic cells treated with AHL are more prone to the formation of biofilms. Another possible approach is to isolate RNA from an earlier time point, where the difference in gene expression between treated and non-treated attached cells could be distinguished. In opposite, conclusion from lines 279-281 should be re-written, as at the moment there is no proof that 3-oxo-C8-AHL positively regulates biofilm production by increasing transcriptional levels of pel genes

Minor comments:

  • Figure 1- how the difference of 29.9 % is calculated? This is not obvious from the graph. No * is signed in the graph as indicated in Figure legend.
  • In the Introduction section, natural production of AHL by ferrooxidans is discussed. This part of introduction can be shortened as it is a broad discussion on authors’ previous work (refs 20- 26). I suggest, if possible, that authors include some comments on QS regulon genes in model organism from this study, A. thiooxidans.
  • line 27 more appropriate term should be used instead of transcriptomics as in this study only several genes were analyzed
  • line 63 remove double reference

Reviewer 3 Report

Please see the file attached.

Round 2

Reviewer 1 Report

The authors have addressed my concerns for their paper.  There are still a few minor typos to correct in particular the Figure 2 legend has "was determined" twice.

Author Response

Figure 2 legend has been corrected. One copy of "was determined" has been modified.

Author Response

Dear Reviewer,

All the minor revisions that you have pointed out have been done.

Thanks a lot.

Best regards,

Nicolas G.